# Sleep stage prediction using multimodal body network and circadian rhythm

Sahar Waqar[1] and Muhammad Usman Ghani Khan[2]

[1] Department of Computer Engineering, University of Engineering and Technology, Lahore, Lahore, Punjab, Pakistan
[2] Department of Computer Science, University of Engineering and Technology, Lahore, Lahore, Punjab, Pakistan



## ABSTRACT

Quality sleep plays a vital role in living beings as it contributes extensively to the healing process and the removal of waste products from the body. Poor sleep may lead to depression, memory deficits, heart, and metabolic problems, *etc*. Sleep usually works in cycles and repeats itself by transitioning into different stages of sleep. This study is unique in that it uses wearable devices to collect multiple parameters from subjects and uses this information to predict sleep stages and sleep patterns. For the multivariate multiclass sleep stage prediction problem, we have experimented with both memoryless (ML) and memory-based models on seven database instances, that is, five from the collected dataset and two from the existing datasets. The Random Forest classifier outclassed the ML models that are LR, MLP, kNN, and SVM with accuracy (ACC) of 0.96 and Cohen Kappa 0.96, and the memory-based model long short-term memory (LSTM) performed well on all the datasets with the maximum attained accuracy of 0.88 and Kappa 0.82. The proposed methodology was also validated on a longitudinal dataset, the Multiethnic Study of Atherosclerosis (MESA), with ACC and Kappa of 0.75 and 0.64 for ML models and 0.86 and 0.78 for memory-based models, respectively, and from another benchmarked Apple Watch dataset available on Physio-Net with ACC and Kappa of 0.93 and 0.93 for ML and 0.92 and 0.87 for memory-based models, respectively. The given methodology showed better results than the original work and indicates that the memory-based method works better to capture the sleep pattern.

## INTRODUCTION

Sleep is a crucial part of the human growth and healing process and typically takes one third of our life (*Colten, Altevogt & Institute of Medicine (US) Committee on Sleep Medicine and Research, 2006*). Scientists recommend 7 or more hours of sleep for adults, and this range of brackets varies with age groups (*Hirshkowitz et al., 2015*). Sleep deprivation has been found to result in the readjustment of a person's sleep cycle and biological clock and cause a compromised immune system, cardiovascular diseases, metabolic disorders, diabetes, mental disorders, depression, anxiety, low impulse control, hallucinations, slackening, reduced decision-making, thinking and memory capabilities (*He et al., 2017*), death (*Cappuccio et al., 2010*), *etc*. Sleep deprivation could be the result of

Corresponding author
Sahar Waqar,
sahar.waqar@uet.edu.pk

intentional delays or certain kinds of diseases like schizophrenia, Alzheimer's, cancer, stroke, stress, aging, sleep apnea, chronic pain syndrome, *etc.*

Quantity and quality contribute to the sleep assessment. Increasing the quantity does not imply better quality. The quantity can be measured easily by tracking the number of hours, but the quality measurement is not straightforward. It defines how early you fell asleep after going to bed, how often you woke up at night, felt rested and energized (*Pilcher, Ginter & Sadowsky, 1997*), and the transitioning and distribution of sleep, stages, *etc.*

To monitor subjects with suspected sleep deprivation, it is important to accurately classify sleep stages. In 2007, the sleep stages were reclassified into four stages (previously five; *Colten, Altevogt & Institute of Medicine (US) Committee on Sleep Medicine and Research, 2006*) according to the American Academy of Sleep Medicine (AASM) standard (*Novelli, Ferri & Bruni, 2010*) which includes three stages in non-rapid eye movement (NREM) and rapid eye movement (REM) sleep with a normal length of 1–5, 10–60, 20–40 and 10–60 min, respectively. They alternate clinically during a sleep episode (*Carskadon & Dement, 2010*). The former is associated with psychological activity and low muscle tone (muscle tension), while in the latter EEG is asynchronous, muscles are atonic (loss of muscle tension) and dreaming is common (*Carskadon & Dement, 2010*). In stage 1 (N1), the body relaxes, the brain activity slows down, and the person can easily awake to noise. Alpha waves are observed at this stage (*Carskadon & Dement, 2010*). These waves are seen in the electroencephalogram (EEG) patterns when a person is in a wakeful state and is quietly resting (*Berry & Wagner, 2014*). In stage 2 (N2), body temperature drops, eye movements stop, and heart rate (HR) and breathing slow down. A short burst of activities can be observed which helps resist external stimuli to avoid waking up. Half of the time in sleep is spent at this stage. The K-complex and sleep spindles can be observed in the EEG pattern (*Carskadon & Dement, 2010*). In stage 3 (N3), deep sleep prevails. A special pattern of delta waves can be observed in this stage. This stage contributes to body healing, recovery, growth, memory creation, organization, insightful thinking, *etc.* Stage 4 (N4) has been merged with N3 in the AASM standard. They collectively create 'slow wave sleep'. These have the highest arousal threshold. Finally, in REM sleep, there is desynchronized low-voltage brain activity that is comparable to that of an awake person and bursts of rapid eye movements (*Carskadon & Dement, 2010*). Muscles (except for the eye and breathing) are paralyzed to avoid any physical movement or fatal accident. Dreaming is identified with an uptick in brain waves. This stage contributes to the development of cognitive skills, learning, and memory functions (*Crick & Mitchison, 1983*). Difference in HR, brain activity, temperature, sympathetic nervous systems, *etc.* in NREM and REM sleep are discussed in (*Madsen et al., 1991*; *Somers et al., 1993*).

The analysis of these stages helps in the assessment of sleep quality. NREM and REM sleep alternate cyclically during a sleep episode (*Colten, Altevogt & Institute of Medicine (US) Committee on Sleep Medicine and Research, 2006*). There can be four to six sleep episodes/cycles of possibly variable lengths with an average of 90 min each. The first cycle is usually the shortest (*Novelli, Ferri & Bruni, 2010*). Patients who have sleep disorders may have no sleep stages and irregular cycles, that is, they may enter REM sleep directly instead

of NREM first (*Keenan, Hirshkowitz & Casseres, 2013*). Therefore, to measure the sleep score, it is necessary to measure the cycle of sleep stages and their transition.

To measure sleep quality/score/stages, different methods such as visual polysomnography (PSG) (*Yildirim, Baloglu & Acharya, 2019*), EEG (*Eldele et al., 2021*; *Michielli, Acharya & Molinari, 2019*), and wrist wearable devices (*Moser et al., 2009*) are used. The former (PSG, EEG) is time-consuming, uncomfortable, and dependent on the experimentation setup in the sleep labs and on the bias/subjectivity of the expert measuring it. Additionally, a large set of EEG features (*Zhu, Luo & Yu, 2020*) and automated feature calculation from convolutional neural network (CNN) models (*Moser et al., 2009*) add complexity to the feature and modeling layers. A handful of studies exist that use the latter approach using devices such as ActiGraph (*Moser et al., 2009*), Apple Watch (*Walch et al., 2019*), *etc*. However, the results are not promising due to the increased sensitivity of wearable devices to noise (*Moser et al., 2009*), the number of classes, unbalanced data (*Eldele et al., 2021*), lack of validation, and, the temporal context.

The use of smartwatches has been well received by the public in the last decade. Wearable devices provide information related to HR, activity, sleep stages, ECG patterns, *etc*. These devices also assess the sleep quality of subjects. However, there is a stall among the research community in their use due to the validity of the algorithms used to calculate the healthcare parameters.

The need was felt to use noninvasive benchmarked devices (like Fitbit and Apple Watch, *etc*.) and study the role of muscle movement, heart activity, and biological clock for sleep stage classification, understand or predict sleep stage transition based on these features and validate it using the PSG study. The results indicate that memory-based models can be used to understand the temporal context of sleep stages and forecast sleep patterns based on the information available from wearable devices. Random Forest (RF), k-nearest neighbors (kNN), and long short-term memory (LSTM) showed better results for multiclass sleep stage prediction problems than existing studies (*Walch et al., 2019*).

The study was carried out to understand the potential of consumer-based wearable devices in the healthcare sector. The idea was to acquire as much information as possible from the device and use it in the diagnosis of diseases and as an early prevention measure. This study can be differentiated from the rest on the following merits:

- A data extraction and storage (Datalayer) model was created that extracted all the information from the consumer-based wearable device (Fitbit Versa 3). This framework is capable of extracting per-interval (second/minute) data from the Fitbit servers from the provided subject ids. This includes heart rate, heart rate variability, elevation, distance, steps, activity modes and levels, $SPO_2$, temperature, sleep stages, food, nutrition, *etc*. As per the literature, no data extraction layer (DAL) has been created for sleep pattern analysis (other than Fitbit's original database—private).
- This study illustrates the dependence of historical time series events on the current sleep cycle. The episodic and repetitive nature of sleep is evident in a sleep pattern. This has not been addressed in the literature.

- The multivariate multiclass sleep stage classification problem has not generated promising results for more classes using wearable devices. Other studies use ECG or EEG *etc.* for collecting data. The results discussed in the study are better than the existing ones.
- The current limited features set and the proposed model showed better results for four and six classes.
- The proposed model is simple with few layers, thus reducing the complexity.

The document is distributed as follows: "Dataset, Preprocessing, and Feature Extraction" discusses the datasets. "Methodology" addresses the models (and their results) that are used for the prediction of sleep stages. Finally, the conclusion is addressed in "Conclusion".

## LITERATURE SURVEY

Sleep has been analyzed by different authors for multiple contexts, that is, to avoid accidents during sleepwalking (*Damkliang et al., 2019*) or on roads (*Chowdhury et al., 2019*; *Patrick et al., 2016*) to understand sleep behavior and patterns (*Budak et al., 2019*; *Hunter et al., 2021*; *Zhang et al., 2022*), to measure sleep quality (*Hunter et al., 2021*), detection of sleep stages (*Gaiduk et al., 2018*), related diseases (*Mitsukura et al., 2020*; *Zhang et al., 2022*), *etc.* For example, *Damkliang et al. (2019)* worked on the detection of the sleepwalking algorithm with three classes (No, Slow, Quick) that were part of the awake state of sleep. For this, they used Samsung Gear Fit smart watches to measure activity data and information coming from the activity sensor. They assigned classes based on 2-min activity. For verification, the authors generated multiple contradictory and actual scenarios like moving oneself without walking, walking in practice, *etc.* These were corrected and identified by the algorithm.

In another study, it was observed that cardiac rhythm can be used to assess sleep quality. For example, *Mitsukura et al. (2020)* only used ECG readings to calculate Heart Rate Variability (HRV) and predicted four and five stages of sleep with an accuracy of around 70% and 66%, respectively. The results were generated using Recurrent Neural Networks (highest), hidden Markov model, and support vector machines, *etc. Sridhar et al. (2020)* uses electrocardiogram (ECG) and extracts Instantaneous Heart Rate (IHR) time-series data from over 10,000 nights of data from the Sleep Heart Health Study (SHHS) and the Multi-Ethnic Study of Atherosclerosis (MESA) with an overall performance of 0.77 accuracy with four classes for every 30 s of sleep. Similarly, *Penzel et al. (2016)* uses cardiac signals, that is, ECG and HR, for the classification of sleep stages, and *Gaiduk et al. (2018)* used HR, activity, and an additional respiratory signal to classify sleep stages with Cohen Kappa of 0.67 and 0.53 for three and four stages, respectively, while *Gaiduk et al. (2019)* used the same sensors for the classification of sleep/wake stages with accuracy 84% and Cohen Kappa of 0.44. They mathematically modeled the features (extracted from the Charite Clinic in Berlin) and used them with multinomial linear regression (MLR) modeling. The results showed that REM sleep was confused with the wake stage due to similar breathing patterns and data imbalance (*Gaiduk et al., 2018*).

For improving the results, recently the authors began to incline toward the use of neural networks. For example, deep learning models, *i.e.*, U-Net, were used to classify a total of seven classes (five sleep stages, and apnea/arousal) using PSG recordings which were auto-annotated using the model. The model was tested on PhysioNet (*Goldberger et al., 2000*) and validated using the SHHS-1 dataset. The results were quite promising with an accuracy AUROC of 0.9826–0.8913 (*Zhang et al., 2022*). However, the N1 stage was confused or misclassified with the N2 stage. Convolution neural network (CNN) and long short-term memory (LSTM) have also been considered in several studies to capture the history of sleep cycles. PSG recordings were used for the detection of four and five sleep stages with an accuracy of 55% and 40%, respectively (*Stuburic, Gaiduk & Seepold, 2020*). Interestingly, a similar study was conducted with cows using neck muscle activity and heart rate to understand their sleep patterns. Dataset collection was a challenge, and the same data imbalance issue persisted. The results showed the classification accuracy of 82% both for RF and neural network (*Hunter et al., 2021*).

Ultimately, all existing studies faced a few common problems: (1) Data imbalance which caused misclassification in higher transitional stages such as REM and N3/N2, *etc*. (2) As a consequence of (1), misclassification of one or two stages as the other stage, causing a decrease in accuracy. Another observation is that the literature is inclined toward (1) the use of PSG recordings or data which is a physical inconvenience for the patient/subject. (2) Dependence on ECG on EEG multilead recordings for feature extraction. (3) Use of complex models due to multilayer networks or a large set of features.

In our study, we have tried to alleviate the above-mentioned issues by using the smart wearable device (Fitbit Versa 3) to collect features and reduce physical inconvenience. Data was collected using minute intervals and 30 s for the MESA dataset. We have augmented or interpolated the data for fewer entries with more subintervals rather than one big interval (increasing the data/data augmentation). The collected data contained heart rate, Heart Rate Variability (HRV), activity, oxygen saturation, *etc*. as features. The activity is also added using the burned calories. We also used the natural class imbalance to capture the time-series pattern in the data stream using a memory-based multivariate model. A simpler context-aware LSTM model was used to capture the historical sleep pattern. The results of Fitbit for the sleep stage prediction problem should be compared with some standard or medical device (*Liang & Chapa-Martell, 2019*). It was noted that the device performance is satisfactory in measuring sleep efficiency, total sleep time (TST), transition probabilities from light to REM, deep to wake, and staying in REM. All other transition probabilities deviated from those collected from the medical device (*Liang & Chapa-Martell, 2019*). For this, we validated our results with the MESA and PhysioNet dataset.

## DATASET, PREPROCESSING, AND FEATURE EXTRACTION

In our study, we utilized the MESA and Apple Watch datasets as validation datasets to assess the generalizability of proposed methdology. These datasets were chosen for their broad demographic coverage and richness in sleep-related data. Complementing them, we collected a detailed dataset in a controlled environment, focusing specifically on local

demographic. Subsequent sections explains the process of custom dataset along with preprocessing and usage of exisiting dataset for the methodology.

## The collected dataset

The device collects different parameters such as activity, nutrition, HR, steps, oxygen saturation (SPO2), respiration rate (RR), HRV, *etc*. Some of these are based on time-series data, while others are measured singularly per day. Subjects were asked to wear the device for up to three months so that sufficient reliable information could be collected. *HR*, sleep stages, and steps were collected per minute while *SPO*2, *RR*, temperature, and other parameters were collected according to the availability of the data. The reception of the data may be affected by the level of battery life and the wearing state of the device. Users were asked to keep the battery of Fitbit at stable levels and sync the data on Fitbit servers regularly. A data framework was developed that sends requests to FitBit servers using APIs 1.1 and 1.2 (provided by Fitbit), receives the response, parses it, and saves it to relational databases using the library *pyodbc* and MS SQL SERVER 2018. Given the variety of data types provided by the wearable device, a data model was created for the time series dataset. A client secret, API key, date of start of data collection, age, and other parameters were recorded in the databases to keep track of multiple patients.

This sampling rate of per minute for sleep stages is considered sufficient to accurately track the various stages of sleep, including light, deep, and REM sleep, which are critical for a comprehensive analysis of sleep patterns. Human sleep cycles, which last approximately 90 min (*Suni, 2023*; *Medical News Today, 2023*; *Penzel, 1999*), do not exhibit rapid changes minute-to-minute, thus making a 1-min sampling interval appropriate for capturing significant transitions without unnecessary detail. Furthermore, this rate is a balanced tradeoff that considers device limitations such as battery life and data storage capacity. Sampling more frequently would yield more granular data but at the cost of increased power consumption and data overload, which could be impractical for continuous overnight monitoring. Additionally, it ensures that the data processing remains within the capabilities of wearable devices, which often have limited processing power. Hence, the 60-s sampling rate is deemed a well-considered approach that balances the need for detailed and accurate sleep analysis with the practical limitations and user-friendliness of sleep tracking devices.

### Features

Heart rate was collected per minute. Information related to HRV and heart zones were also collected. The zone provides min and max HR, and the calories burned for a particular type of heart activity such as cardio, fat burn, peak, *etc*.

Activity data is based on the accelerometer and gyroscope sensor readings, *etc*. However, Fitbit does not provide raw acceleration in three dimensions. Instead, it provides time-series data for steps, elevation, distance, calories, and sedentary, light, fair-active, and very active per minute classes.

Sleep: Fitbit provides two APIs for the extraction of sleep data. API 1.1 provides time-series data for three stages, *i.e.*, awake, restless, and sleep, while API 1.2 provides

time-series data for four stages, *i.e.*, wake, light, deep, and REM sleep. Both types of data were collected using a data framework. All the entries were collected using 60 s intervals. However, Fitbit provides long and short sleep cycles. Short cycles are also stored using 30 s intervals. Therefore, after data collection, the Sleep Stage Tables contained entries for both 30 s and per-minute intervals (causing duplicate entries). All were later interpolated and set to one interval.

Temperature: Fitbit also provides temperature per minute. This was also captured and saved.

Other than all the above-mentioned items, information related to the food journal, body BMI, weight, water intake, age, gender, medical conditions, allergies, snore, RR, *etc.* was also collected.

### Data preprocessing

**Interval mapping:** For data preparation, all tables were naturally joined based on the same time. The information in the HR and Sleep Interpolated tables had 190,688 ($m$) and approximately 67,944 ($n$) entries. After joining, $min(m, n)$ rows were produced. This was because sleep was measured during the night only while other activities (HR, distance, steps, *etc.*) were measured during daylight as well. Therefore, entries containing both sleep and activity information during night hours were considered (67,944).

Moreover, some of the HR and sleep entries have different measurement time instances. For example, HR was measured at 6:00:00, while sleep activity was at 6:00:30. Such a row will not appear in the natural join. It was resolved by taking the floor of the sleep time instants. It was done carefully assuming that during 30 s intervals, the sleep stage will not change.

**Interpolation/data filling:** As explained earlier, Fitbit provides two types of data in API 1.2 for sleep stages *i.e.*, short, and long sleep cycles. It provides the start time of the sleep stage, name, type (short and long), and total time spent in that stage. These entries could not be joined with the existing time series tables that contained per-minute entries. Therefore, we converted the summarized information to expanded time-based data. The minimum interval of the sleep entries was 30 s for both long and short cycles. To avoid loss of data, the sleep tables were updated for 30 s intervals. For example, for an entry of a subject's sleep at 6:30:00 in deep sleep for the 1,200 s ($1,200/60 = 20 \, min$), 40 new entries (with 30 s interval: $20 * 2$) were created with the class as the deep sleep stage and the time from 6:30:00 to 6:50:00. Furthermore, data from short sleep cycles were merged with long sleep cycles by placing them in their proper location (sorted by time). For conflict times (duplicate data) between short and long sleep cycles, as shown in Table 1, intervals were replaced with the short interval data. After this, the dataset was prepared and merged with the other per-minute data tables and stored in the SQL database. A total of $n$ records were created from 31,867 records. Some samples have been shown in Table 1.

Data for sleep stages from API 1.2 was unbalanced. For a total of $n$ rows, the wake class consisted of 12.2%, light class 50.1%, deep class 18.1%, and REM class 19.6%. This data imbalance is natural because humans, as described above, spend more time in light sleep

**Table 1  Time series sleep stage data with duplicates.**

| Timestamp | Interval | Stage | Type |
|---|---|---|---|
| 2021-12-16 06:37:00 | 30 | Light | Long |
| 2021-12-16 06:37:30 | 30 | Light | Long |
| 2021-12-16 06:38:00 | 60 | Wake | Short |
| 2021-12-16 06:38:00 | 30 | Light | Long |
| 2021-12-16 06:38:30 | 30 | Light | Long |
| 2021-12-16 06:38:30 | 30 | Wake | Short |

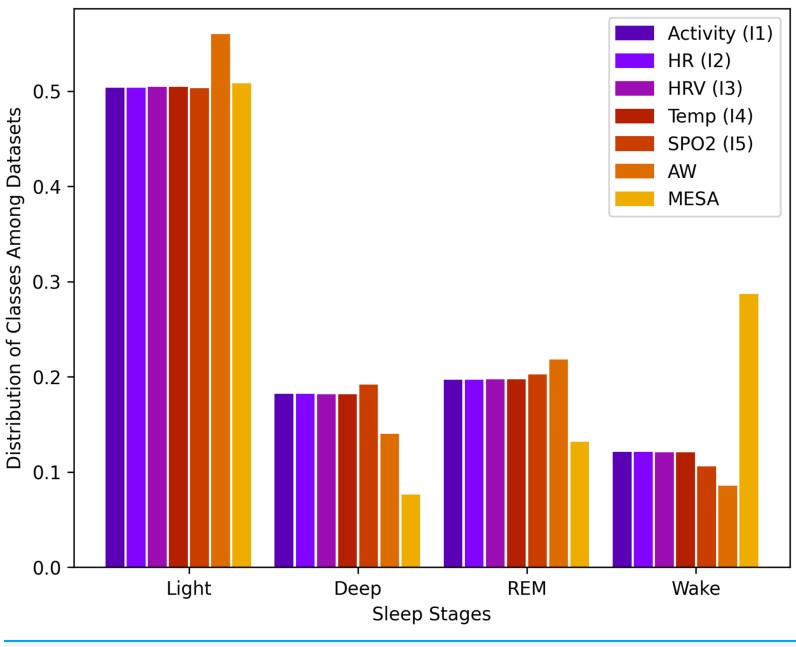

**Figure 1  Sleep stage class distribution.**

than in the other stages of sleep. This distribution of stages, along with all other instances, is shown in Fig. 1.

**Dataset creation:** Finally, all tables were merged to create the following database instances as shown in Table 2. The high-level attributes also contain the patientId, and the class attribute (sleep stage). Different instances were created to observe the impact of each parameter on the prediction of the sleep stage. It should be noted here that the extracted information depends on the Fitbit wearable device. So, some tables contained fewer entries than others.

## Validation dataset

To validate our results, we have used both expert predictions from the Fitbit device and tested our methodology on another benchmark study and a longitudinal study of the Multi-Ethnic Study of Atherosclerosis (MESA).

**Table 2 Database instances.**

| Name | High level attributes | Entries | Features | Interval |
|---|---|---|---|---|
| $I1_{act}$ | PatientId, Time, Activity (Levels, Mets, Calories, Distance, Elevation, Floors, FairActive, LightActive, veryActive, Sedentary, Steps), and Sleep Stages | 33,971 | 14 | 60 |
| $I2_{hr}$ | Features of $I1_{act}$, HR, and $HRV_{calc}$ | 33,971 | 16 | 60 |
| $I3_{hrv}$ | Features of $I2_{hr}$, HRV (rmssd, coverage, $f_{high}$, and $f_{low}$) | 33,739 | 19 | 60 |
| $I4_{temp}$ | Features of $I3_{hrv}$ and Temperature | 33,739 | 20 | 60 |
| $I5_{spo2}$ | Features of $I4_{temp}$ and Oxygen ($SpO_2$) | 10,890 | 21 | 60 |
| AW | PatientId, Activity (steps), Time (Time, Circadian, Cosine), HR, and Sleep Stages | 25,482 | 7 | 30 |
| MESA | PatientId, Activity (steps), Time (Time, Circadian, Cosine), HR, and Sleep Stages | 942,012 | 7 | 15 |

### Apple Watch PhysioBank dataset

In this dataset HR, raw acceleration, steps, and circadian rhythm were collected from 39 subjects using the Apple Watch for some days. In the end, they spent the night in a sleep lab for polysomnography (PSG) for an 8-h sleep. The readings were recorded and their sleep stages were labeled by the experts, tagged, and saved (*Goldberger et al., 2000*; *Walch et al., 2019*).

### The Multi-Ethnic Study of Atherosclerosis dataset (MESA)

It is a longitudinal study for the investigation of subclinical to clinical cardiovascular disease (CVD) in a multi-ethnic community. Subjects enrolled in the study had four follow-up exams from 2003 to 2011. In 2010, 2,237 participants were recruited for the Sleep Exam, *i.e.*, full unattended PSG overnight and 7-day actigraphy to understand sleep and its disorders among different ethnic groups and related to subclinical atherosclerosis (*Chen et al., 2015*; *Zhang et al., 2018*). Multiple studies have used this dataset to validate their work (*Perez-Pozuelo et al., 2022*; *Tang et al., 2022*; *Walch et al., 2019*) for the classification of the sleep stage. Using *Walch et al. (2019)*, the same method was used to extract the features from MESA as of the Apple Watch dataset to make it comparable.

## Data preparation

An identical set of features was extracted from both the Apple Watch and MESA datasets using the same technique.

### Feature preparation

- The raw acceleration $(x, y, z)$ was converted to activity counts using the technique by Lindert (*Te Lindert & Van Someren, 2013*) and convolved with Guassian ($\sigma = 50$) (*Walch et al., 2019*).
- The heart rate was augmented for a 1-s interval and smoothened. Later, it was convolved with a difference of the Gaussian filter ($\sigma = 120\,s, \sigma = 600\,s$) to highlight the periods with change.

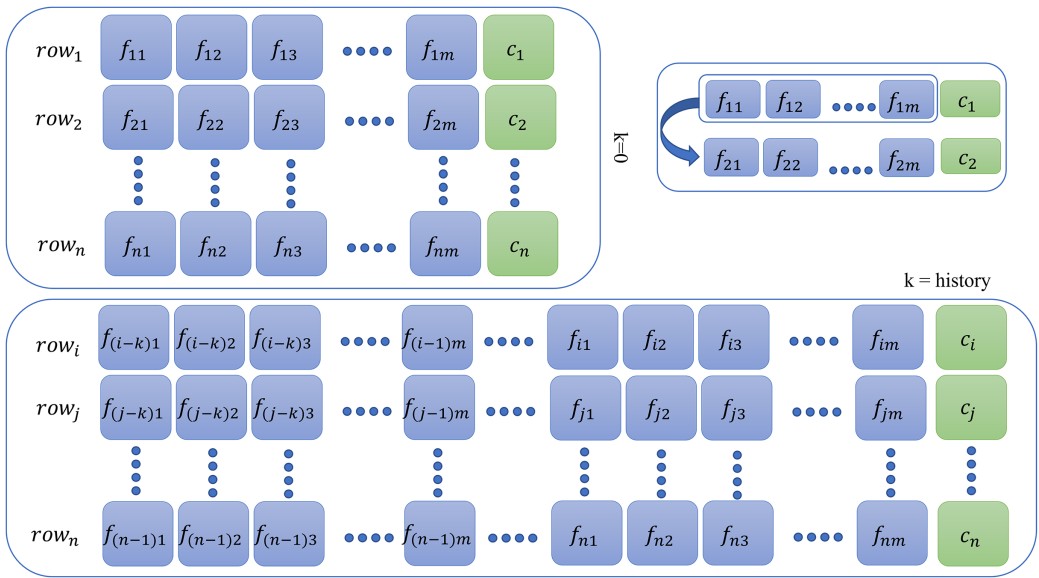

**Figure 2 Context preparation for time series data.**

- Heart rate was also normalized by taking the absolute difference of consecutive heart rate divided by the mean of the feature over a sleep period (*Walch et al., 2019*). This gives the variation in heart rate.

- The clock $C$ was modeled using the cosine wave that rises and falls overnight with respect to time $t$ as given by Eq. (1).

$$C = cos(t) \tag{1}$$

- Circadian drive refers to the biological clock that drives the need for sleep and awakening. It is modelled using the circadian clock model given by the following Eqs. (2) and (3) proposed in *Forger, Jewett & Kronauer (1999)* where $B$ is the effect of light, $\tau_x$ is the oscillator period, $kB$ is used to adjust the direct effect of light, $x$ is the initial condition, and $\mu = 0.23$ (best fit). This model captures the effect of light on the generation of the circadian rhythm of the human body. Inherently, it uses the classic van der Pol oscillator (model with cubic nonlinearity) with Process L and Jewwet and Kronauer's model of Aschoff's rule. This model (*Forger, Jewett & Kronauer, 1999*) is simpler than the rest (non-linearity with degree 7).

$$\frac{dx}{dt} = \frac{\pi}{12}(x_c + B) \tag{2}$$

$$\frac{dx_c}{dt} = \frac{\pi}{12}\left[\mu\left(x_c - \frac{4x_c^3}{3}\right) - x\left(\frac{24}{0.99669\tau_x}\right)^2 + kB\right] \tag{3}$$

Wearable devices do not provide light information. Therefore, the activity data were transformed into light information with the notion that activity is usually carried out under light conditions, as suggested in *Walch et al. (2019)*. It is important to note that light

information serves as the context for the activities being carried out. This context is maintained in the self-collected dataset, while to have consistent data across the datasets, this measure serves as empirical approximation to infer the light information from the activities.

### Sequence generation

After feature calculation, the sliding window method (SWM) was used to prepare the final data. The sliding window technique is crucial for preparing time series data, ensuring consistent input dimensions by enhancing the model's ability to generalize from training on overlapping data segments. This method is adaptable to different data intervals, facilitates real-time learning, and optimizes the use of data, thereby markedly enhancing the efficacy and precision of predictive models.

This additional step is required for memory-based models only for all database instances. For context or history learning, we need to readjust the previous *rows* (containing features at the time $i$) of the data to $row\prime$ which contains information of previous $n$ time instants, that is, if $n = 2$ (two steps back), then all rows of the new dataset will contain entries as given by the following sequence: $row_{i-2}, row_{i-1}, row_i, class$ as shown in Fig. 2. For memory-based models, each database instance was regenerated $1-40$ times ($n$) using SWM. Thus, depending on the value of $n$, that is, $k = instances * n$, datasets were created for experimentation purposes. The larger the $n$, the more complex the model becomes and requires a tremendous amount of memory and time resources. The Results section only illustrates the result of the best-fit $n$ steps that achieved meticulous results. The results for all other history steps and datasets are discussed in the Supplemental Material.

## METHODOLOGY

The study was approved as referred to in letter No. ORIC/110-ASRB/1647 issued by the Advanced Studies and Research Board, Office of Research, Innovation and Commercialization, University of Engineering and Technology Lahore, Pakistan. The dataset for this research activity was collected using the Fitbit Versa 3 wearable device in two subjects (one male and one female) with their written informed consent. The problem at hand is time-series data. As sleep progresses through the night, certain patterns repeat themselves. Usually, each cycle is around 90 min long. This indicates that the data has certain kind of information which is dependent on past events, *i.e.*, if a subject at time $t$ with HR 68, and 0 muscle tone is in stage 2, then he may transition to another stage at time $t + 1$ with the same or different parameters. The idea is to capture this pattern that remembers the history before the prediction.

For this particular study, we have used both machine learning (memory-less) and time-series models to see the effect of history on sleep stage prediction and forecasting. This idea was further extended to generate a sleep pattern and compared with the original sleep pattern. Overall methodlogy is shown in Fig. 3.

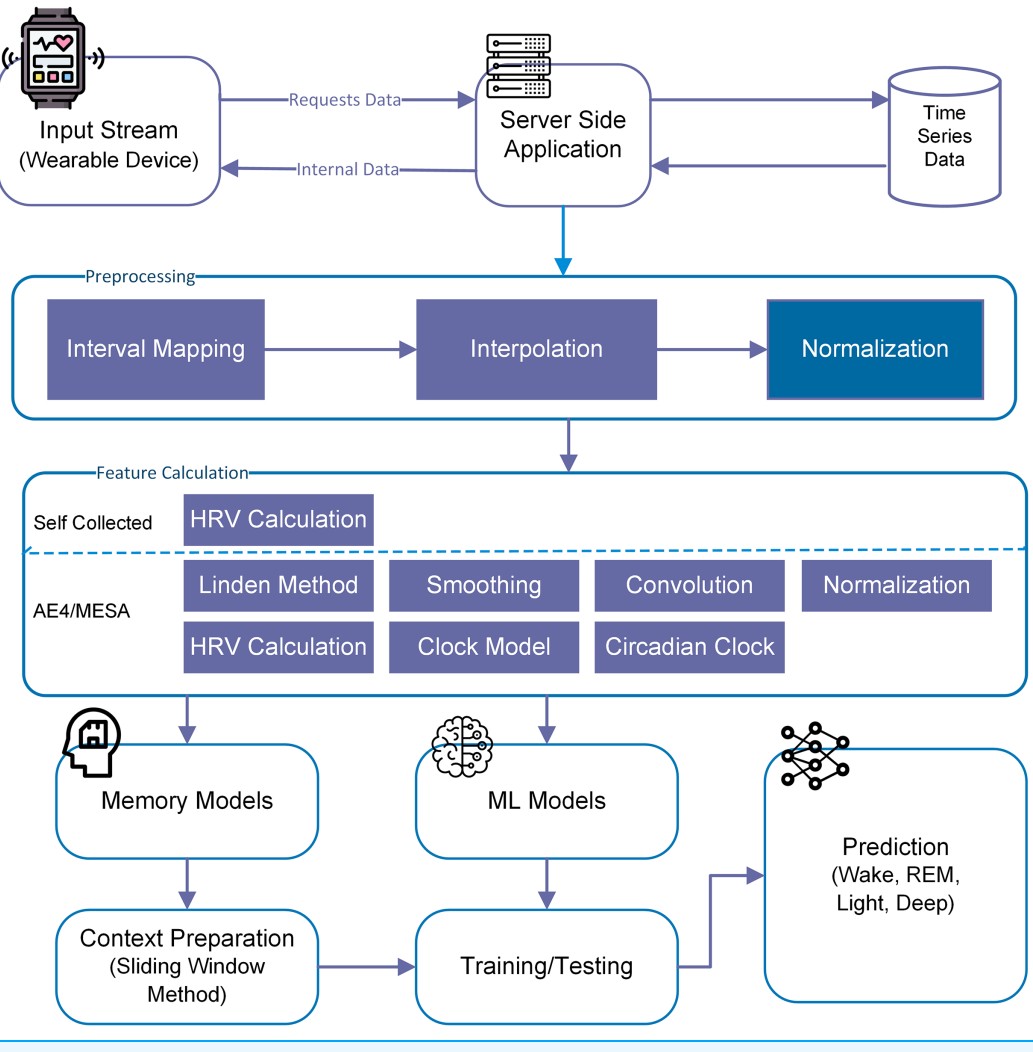

**Figure 3** Overall methodology for sleep stage prediction.

## Machine learning (ML) models

For four sleep stages prediction, we have tested all DB instances as represented in Table 2 with logistic regression (LR), Random Forest (RF), k-nearest neighbors (kNN), support vector machine (SVM, $kernel = rbf$) and multi-layer perceptron (MLP) and observed the effect of each additional feature on the prediction performance. GridSearchCV was used to test 10–1,000 models, and most optimal parameters were used to report the results. The subject-independent stratified train and test sets were divided with 0.75 and 0.25 ratios, respectively. The characteristics of each instance are illustrated in Table 2.

### Results

Comparison of the proposed technique is along with results is shown in Table 3 while the results of the models are shown in Tables 4 and 5 for four and six classes, respectively, and are also presented in Fig. 4. It is important to note here that due to the imbalance of dataset classes, the weighted evaluation metrics are used. As in a balanced dataset, each class

**Table 3 The comparison of the results with the previous work.**

| Reference | Type | Sensor | Model | Results (ACC) | Dataset | Comments |
|---|---|---|---|---|---|---|
| *Cho et al. (2019)* | 2-stage | Accelerometer (accl) | CNN-LSTM | 88.77 | Self | Tested accl for 2-classes only |
| *Altini & Kinnunen (2021)* | 2-stage | PSG, accl, circadian | Light gradient boosting | 96 | Self | Data validity |
| *Gargees et al. (2019)* | 3-stage | Hydraulic bed sensor | CNN-LSTM | Avg. = 90 | Self | – |
| *Reimer et al. (2018)* | 3-stage | Accl | RF | 86–90 | Self | N1 confused |
| *Reimer et al. (2017)* | 3-stage | Accl | RF | 90–91 (Multi-class) | Self | N1 and N2 confused |
| *Zhai et al. (2020)* | 3-stage | HR/HRV | CNN-LSTM | 78.3, F1 = 69.8 | MESA | The dataset used is collected in lab settings |
| *Yoon et al. (2020)* | 4-stage | Microwave sensor, infrared sensor | kNN | 98.65 | Self | No data available. Difficult to reproduce the results |
| *Reimer et al. (2018)* | 4-stage | Two accl(s) | RF | 86–90 (Multi-class) | Self | Too many features |
| *Hsieh et al. (2021)* | 4-stage | EEG and EOG | DL | 86.72 | Self | Less number of adults in the acquired dataset. |
| *Zhang & Guan (2017)* | 4-stage | EEG | DL | 85.5 | Self | Classes confused. |
| *Zhai et al. (2020)* | 4-stage | ECG, HR, accl | LSTM | 80.75 | – | Poor accuracy in deep stages |
| *Yildirim, Baloglu & Acharya (2019)* | 4 stage | PSG/EEG | CNN | Max (EDF) 92.6, EDFx: 92.33 | SleepEDF and SleepEDFx | N1 was misclassified |
| *Moser et al. (2009)* | 4-stage | Accl, ECG/EKG, SkinTemp | Bagging with DT | 0.71 | Self | – |
| *Altini & Kinnunen (2021)* | 4-stage | PSG, accl, circadian | Light gradient boosting | 79 | Self | Variations in data collection |
| Ours | 4-stage | Act, hr,.., circadian | ML and LSTM | Max. = 93, F1 = 93 | Self, AW, and MESA | Better F1-score |
| *Zhang & Wu (2018)* | 5-stage | PSG/EEG | CNN | 87 | UCD and MIT-BIH | N1-confused |
| *Nakamura et al. (2020)* | 5-stage | EEG | – | 74.1 | Self | N1-confused |
| *Koushik, Amores & Maes (2019)* | 5-stage | EEG | CNN | ACC = – | – | – |
| *Zhu, Luo & Yu (2020)* | 5-stage | PSG/EEG | CNN | EDF overall ACC = 93.7, EDFx ACC = 82.8 | SleepEDF and SleepEDFx | N1 was misclassified |
| *Jadhav et al. (2020)* | 5-stage | EEG | Transfer learning with CNN | Max ACC = 83.34 | SleepEDFx | – |
| *Yildirim, Baloglu & Acharya (2019)* | 6 stage | PSG/EEG | CNN | Max EDF ACC = 91, EDFx: ACC = 89.54 | SleepEDF and SleepEDFx | N1 was misclassified |
| Ours | 6-stage | Act, hr,.., circadian | ML and LSTM | Max. ACC = 88, F1 = 88 | Self, AW, and MESA | Better F1-score |

contributes equally to the overall precision and recall. However, in an imbalanced dataset, where some classes have more samples than others, using standard measures can give a skewed view of the model's performance.

**Table 4 Results for machine learning models on database instances with the four classes.**

| Model | Instance | ACC | PRC | Recall | Kappa | F-score | MC |
|---|---|---|---|---|---|---|---|
| LR | $I1_{act}$ | 50.3 | 100 | 50.3 | 0.0 | 67 | 0.0 |
| | $I2_{hr}$ | 51 | 98 | 51 | 0.0 | 67 | 0.1 |
| | $I3_{hrv}$ | 53 | 80 | 53 | 14 | 62 | 18 |
| | $I4_{temp}$ | 53 | 79 | 53 | 15 | 61 | 19 |
| | $I5_{spo2}$ | 57 | 76 | 57 | 25 | 63 | 29 |
| | AW | 57 | 86 | 57 | 12 | 67 | 16 |
| | MESA | 52 | 80 | 53 | 24 | 61 | 28 |
| MLP | $I1_{act}$ | 50 | 25 | 50 | 0.0 | 34 | 0.0 |
| | $I2_{hr}$ | 54 | 78 | 54 | 18 | 61 | 22 |
| | $I3_{hrv}$ | 62 | 70 | 62 | 40 | 64 | 39 |
| | $I4_{temp}$ | 63 | 70 | 63 | 40 | 65 | 41 |
| | $I5_{spo2}$ | 66 | 72 | 66 | 44 | 67 | 45 |
| | AW | 69 | 74 | 69 | 44 | 70 | 45 |
| | MESA | 54 | 79 | 54 | 28 | 63 | 31 |
| kNN | $I1_{act}$ | 88 | 89 | 88 | 81 | 88 | 81 |
| | $I2_{hr}$ | 61 | 62 | 61 | 41 | 62 | 41 |
| | $I3_{hrv}$ | 78 | 79 | 78 | 67 | 78 | 67 |
| | $I4_{temp}$ | 81 | 82 | 81 | 71 | 81 | 71 |
| | $I5_{spo2}$ | 69 | 69 | 69 | 52 | 69 | 52 |
| | AW | 87 | 87 | 87 | 79 | 87 | 79 |
| | MESA | 73 | 73 | 73 | 62 | 73 | 63 |
| SVM | $I1_{act}$ | 52 | 92 | 53 | 1.0 | 66 | 2.0 |
| | $I2_{hr}$ | 53 | 89 | 54 | 12 | 65 | 19 |
| | $I3_{hrv}$ | 62 | 76 | 61 | 34 | 65 | 38 |
| | $I4_{temp}$ | 63 | 75 | 63 | 37 | 66 | 40 |
| | $I5_{spo2}$ | 67 | 73 | 67 | 45 | 68 | 47 |
| | AW | 66 | 75 | 66 | 38 | 69 | 40 |
| | MESA | – | – | – | – | – | – |
| RF | $I1_{act}$ | 96 | 96 | 96 | 100 | 96 | 94 |
| | $I2_{hr}$ | 73 | 74 | 73 | 58 | 73 | 58 |
| | $I3_{hrv}$ | 88 | 88 | 88 | 82 | 88 | 82 |
| | $I4_{temp}$ | 88 | 88 | 88 | 81 | 88 | 82 |
| | $I5_{spo2}$ | 89 | 89 | 89 | 84 | 89 | 84 |
| | AW | 93 | 93 | 93 | 88 | 93 | 88 |
| | MESA | 75 | 78 | 75 | 64 | 76 | 64 |

It is evident that RF and kNN at $k = 3$ performed more adequately than all other classifiers. RF performed best in $I1_{act}$ and better in $I3_{hrv}$, $I4_{temp}$, $I5_{spo2}$, and $AW$ while kNN performed best in $I1_{act}$ and adequate in $I3_{hrv}$, $I4_{temp}$, and $AW$. This gives rise to the notion that activity or muscle movement plays a vital role in the classification of sleep stages. HR alone does not help in this process because all stages contain approximately all possible HR

**Table 5 Results for the ML models on DB instances with six classes.**

| Model | Instance | ACC | PRC | Recall | Kappa | F-Score | MC |
|-------|----------|-----|-----|--------|-------|---------|-----|
| LR | AW | 51 | 78 | 51 | 14 | 60 | 17 |
| | MESA | 52 | 80 | 53 | 24 | 61 | 28 |
| MLP | AW | 65 | 68 | 65 | 47 | 66 | 47 |
| | MESA | 54 | 79 | 54 | 28 | 63 | 31 |
| kNN | AW | 83 | 83 | 83 | 74 | 83 | 74 |
| | MESA | 73 | 73 | 73 | 62 | 73 | 63 |
| SVM | AW | 63 | 71 | 62 | 40 | 65 | 41 |
| | MESA | – | – | – | – | – | – |
| RF | AW | 88 | 89 | 88 | 82 | 88 | 82 |
| | MESA | 75 | 78 | 75 | 64 | 76 | 64 |

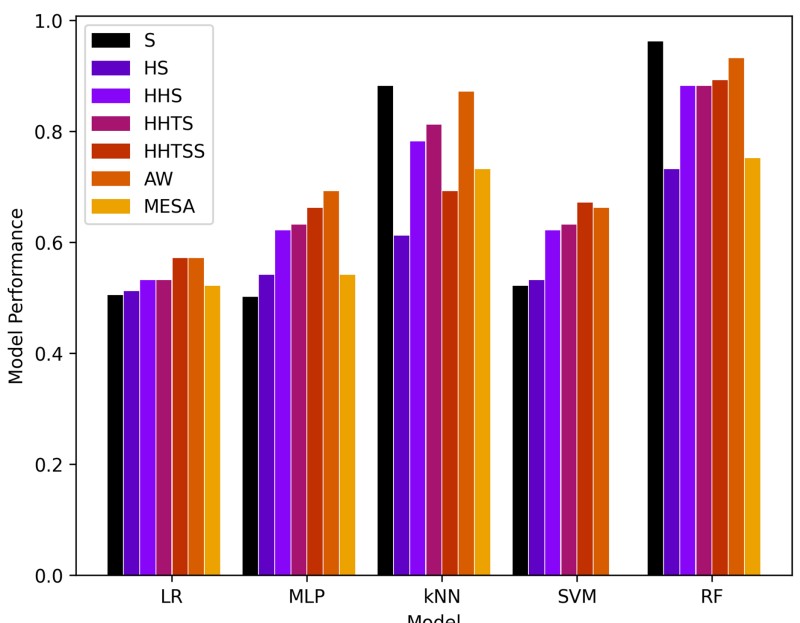

**Figure 4 Results of machine learning based models with the four classes.**

values in a particular normal range of a subject. However, HRV served as a better feature to measure the transition of the sleep stage, as evidenced by an increase in the accuracy in $I3_{hrv}$ from $I2_{hr}$ and a decrease from $I1_{act}$ to $I2_{hr}$ for both RF and kNN. This can also be validated from *AW* and *MESA* instances.

## Comparison with existing studies

Comparing the study with existing work is not straightforward since there is no agreement on how to report the results (*Djanian, Bruun & Nielsen, 2022*).

- Several studies have used accuracy as an evaluation measure. However, with inherent uneven data and multivariate multiclass prediction problems, the accuracy (ACC) does

**Table 6 The comparison of the results with the original work for the Apple Watch dataset.**

| Reference | LR | kNN | RF | Neural Net | LSTM | Train-AW-Test-MESA |
|---|---|---|---|---|---|---|
| Three class (*Walch et al., 2019*) | 0.699 | 0.721 | 0.686 | 0.723 | – | 0.686 |
| Four class proposed | 0.57 | 0.87 | 0.93 | 0.69 | 0.918 | 0.741 |
| Six class proposed | 0.51 | 0.83 | 0.88 | 0.65 | 0.842 | 0.617 |

not serve as a good measure. Mostly F1 score, Cohen Kappa, sensitivity, precision, and confusion matrix have been used.

- Another challenge is the variation in the selection of the training and testing process of the studies. Some of the existing studies are using K-fold cross validation, while others are using leave one out cross validation (LOOCV) (*Djanian, Bruun & Nielsen, 2022*). Some of the studies test their methodologies on different datasets.
- Additionally, some have worked on two-stage, some three-stage, and four-stage while others have on five-stage sleep classification problem.
- There also exists a variation in sensing techniques, *i.e.*, EEG, ECG, PPG, PSG, *etc.*
- There also exists variation in sensing devices for the same sensing type, *i.e.*, PPG from Fitbit device, PPG from Apple Watch, *etc.*
- There exists a variation of the validation dataset. Some have used their own datasets that are not public while others have used PSG datasets that are not validated (*Djanian, Bruun & Nielsen, 2022*).

With the facts listed above, we have tried our best to compare the F1 scores or Cohen Kappa for four and six-class sleep stage studies. Table 6 compares the results with the proposed solution.

In the original work (*Walch et al., 2019*), the best accuracy achieved with the prediction of three sleep stages, namely wake, REM, and NREM, is 0.69 with LR, 0.721 with kNN, 0.686 with RF, and 0.723 with Neural Net as shown in Table 6. Similarly, they validated with MESA and the best-attained accuracy was 0.686 for three class problems. The results achieved in this study were for four and six class problems and better than in the original work. The comparison is shown in Table 6.

## Memory efficient models

The human brain remembers the context and weighs the importance of certain events and ignores irrelevant information accordingly. It is established that sleep usually behaves systematically, is episodic, and is carried out in cycles. Therefore, before making any prediction, it is important to consider the information in the previous time instants *i.e.*, in which stage the subject was present earlier, what was his activity level, or what was the level of the heart activity, *etc.* For this reason, we considered neural networks with a memory like LSTM that have more memory than recurrent neural networks (RNNs). The RNN saves information about the previous state only, while LSTM can remember beyond the recent state. It is also interesting to note the effect of how far to look in the past (*steps*)

before making a decision. With extensive experiments on all possible database instances, we observed that for a certain $k$, results are more optimal than the other, *i.e.*, ..., $k - 2$, $k - 1$, $k + 1$, $k + 2$, ....

### Long short term memory (LSTM)

LSTM has an input gate ($i$) and an internal cell state ($c$), output gate ($o$), input modulation gate ($g$), and forget gate ($f$). Input and output gates take care of the incoming and outgoing while the modulation gate contributes to fineness, and forget gate discards the irrelevant information. $W_q$ and $U_q$ in Eqs. (4)–(7) represents weights of the inputs and recurrent connections, respectively, subscript $q$ the gates, $\sigma_g$ the sigmoid activation function, $\sigma_h$ the hyperbolic tangent function, and $b$ represents the bias.

An internal state can be calculated by the summation of the Hadamard product of the input and modulation gate and the Hadamard product of the previous internal cell state and the forget gate, as shown in the Eq. (8). The current hidden state is ($h_t$) is calculated by the Hadamard product of the output and $c_t$ as shown in Eq. (9). Similarly, using Eqs. (4)–(7) we can calculate the value of the forget state, input, output, and internal cell state at the time $t$.

$$f_t = \sigma_g(W_f x_t + U_f h_{t-1} + b_f) \tag{4}$$

$$i_t = \sigma_g(W_i x_t + U_i h_{t-1} + b_i) \tag{5}$$

$$o_t = \sigma_g(W_o x_t + U_o h_{t-1} + b_o) \tag{6}$$

$$\bar{c}_t = \sigma_g(W_c x_t + U_c h_{t-1} + b_c) \tag{7}$$

$$c_t = i \odot g + f \odot c_{t-1} \tag{8}$$

$$h_t = o_t \odot \sigma_h(c_t) \tag{9}$$

Finally, if $\bar{y}_t$ is the predicted output at time $t$ and $y_t$ is the actual output then the error function for the LSTM is given by the Eq. (10).

$$E_t = -y_t \log(\bar{y}_t) \tag{10}$$

Considering the importance of the forget gate, we experimented with a certain range of memory steps *ns*, starting from 1 to 39 and *epochs* = 50 with sequential LSTM with dense layer, Adam algorithm, and tanh operation. All features were converted to *float*, min-max scaled, or normalized, where required. Negative values were converted to positive scales using the Eq. (11) where $y$ is a column vector. Rows for null cell values were removed. The resultant set was inputted to LSTM. The final architecture used to predict the next sleep stage is shown in Fig. 5.

$$y = \log_2(y + 1 - \min(y)). \tag{11}$$

LSTM was further tested on multiple variants of the database instances. AW and MESA were tested with four (*AW4* and *MESA4*), and six (*AW6* and *MESA6*) classes, training on AW and testing with MEFA (*TAW4TMESA4*, *TAW6TMESA6*), AW and MESA (*Activity + Circadian*) without heart rate (*AW4Act*, *AW6Act*, *MESA4Act*, and *MESA6Act*,

**Peer**J Computer Science

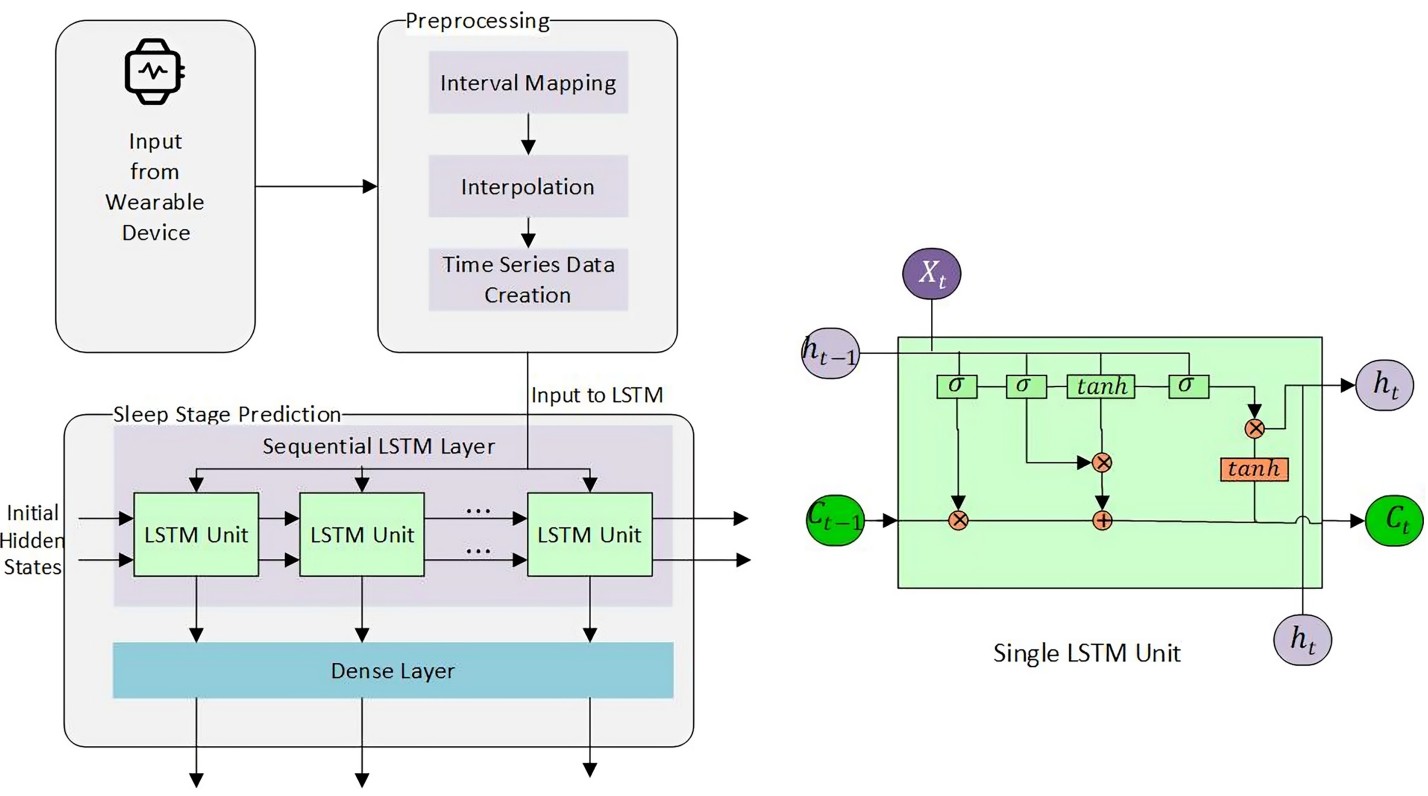

**Figure 5  Sleep stage prediction using LSTM.**

and $TAW4ActTMESAAct$), *etc*. The results for six classes were omitted to limit the scope of the document.

## Results

The number *ns* indicates how far to look back in history and extract a pattern. Instances from $I1_{act}$ to $I5_{spo2}$ were collected in minute intervals, while AW and MESA were collected in 30 and 15 s, respectively. An interesting fact was noted that all instances had their first maximum optimal results below $ns = 15_{spo2}$, as shown in Fig. 6 for Fitbit databases and the validation databases (AW and MESA).

This gives rise to a hypothesis that the next sleep stage can be predicted based on the information of the prior few minutes of activity.

We evaluated all instances using root mean square error (RMSE), Accuracy (ACC), Precision (PRE), Recall (REC), F1 score (F1), Cohen Kappa (CK) and Matthew's constant (MC). The top three results for both the Fitbit and the validation databases are shown in Figs. 7 and 8. It is evident from Fig. 8 that the Apple Watch dataset (AW4) with four classes (wake, light, deep, and REM) had a maximum score for all evaluation parameters, that is, ACC = 0.918 and Cohen = 0.867, *etc*. The training and test loss of this dataset is shown in Fig. 9. This dataset contained information related to circadian rhythm, HRV, and activity. The second best scorer is again AW4 but without HRV, that is, $ACC = 0.902$ and

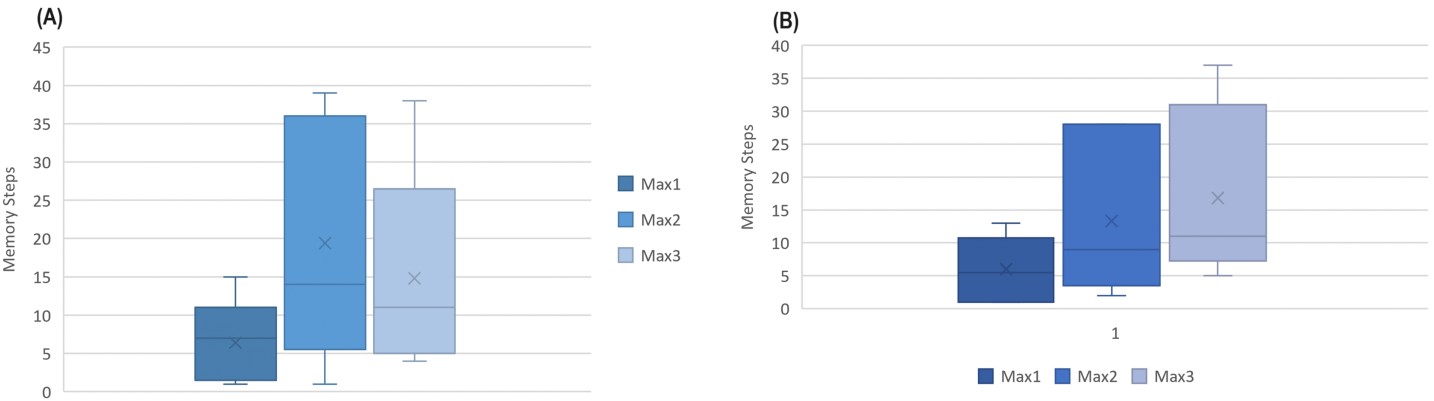

**Figure 6** The boxplot of first three best results (ACC) attained among memory steps 1 to 39 for (A) Fitbit instances and (B) validation instances.

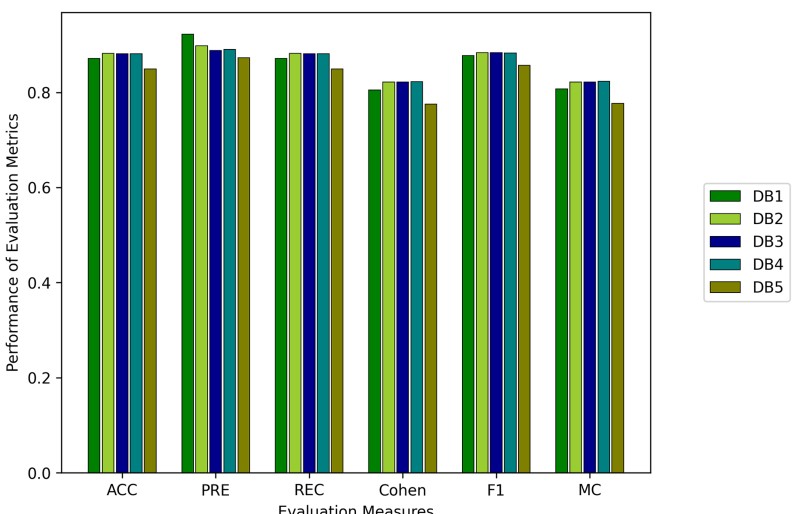

**Figure 7** Results for the fitbit instances using the LSTM model.

Cohen = 0.83, *etc*. The results of both of these instances are comparable and defend the results obtained from the ML algorithms, *i.e.*, activity alone contributes to the prediction of sleep stages and activity with HRV serves as a better measure in sleep stage classification. It is also interesting to note that the same LSTM model was trained on AW and tested on MESA and the results obtained (ACC = 0.783, Cohen = 0.668) were better than the original study used to create this dataset (*Walch et al., 2019*).

**Visualization:** LSTM predicts the complete sequence *s* as given by:

$$s = [circadian, cosine, steps, hr, time, stage].$$

To visually understand the sequence prediction of the proposed LSTM model, we used the Apple Watch's (AW4) testing dataset and used a sleep sequence *s* to predict the sleep

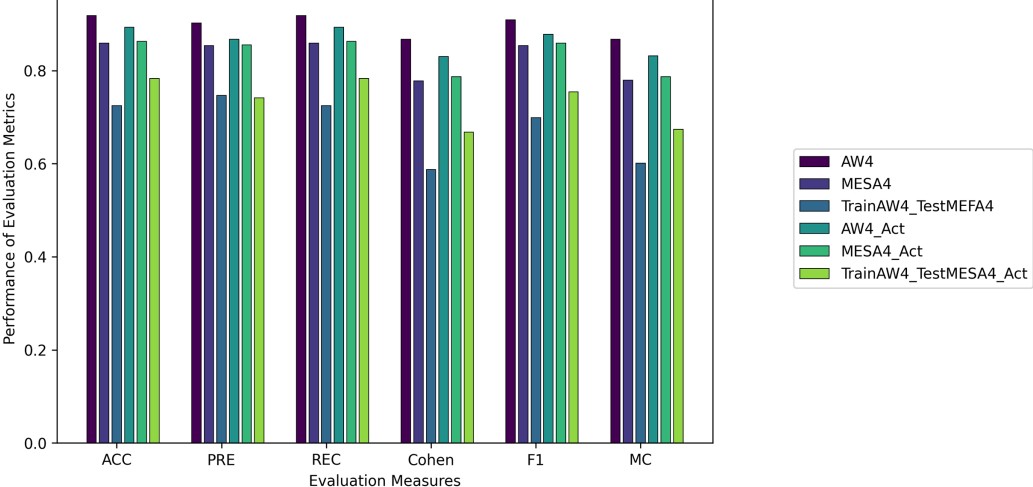

**Figure 8  The results for the validation instances using the LSTM model.**

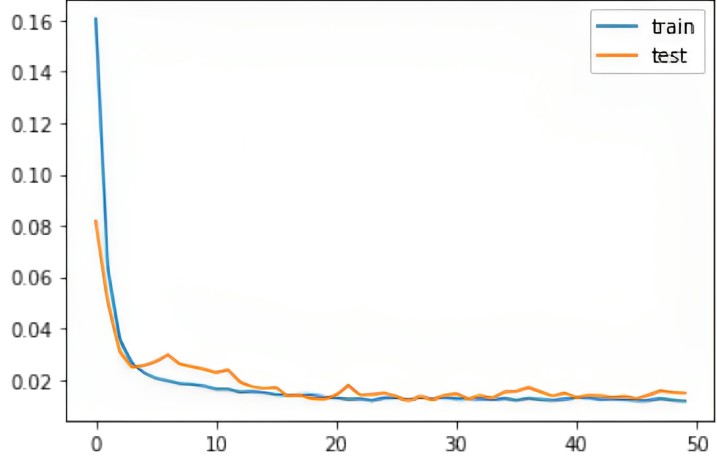

**Figure 9  The training and test RMSE loss for the Apple Watch dataset (AW4).**

pattern. Figure 10 shows the actual circadian rhythm, HR, step, sleep pattern, and the predicted sleep pattern. It can be observed that the proposed model is quite successful in predicting the sleep pattern of the original subject which is comparable with the actual predicted sleep pattern (also validated by the results presented in Fig. 8).

## DISCUSSION

**Memory Models:** The validation datasets $AW4$ and $MESA4$ both attained their first two maximum ACC at $ns = 1$ or $2$ and $ns = 10$ or $11$. The best results were obtained for $ns = 8$ for $AW4Act$ (without HR), for $ns = 7$ for $I2_{hr}$, $I5_{spo2}$, $AW6Act$, and $MESA6Act$ and the second best results for $MESA4Act$ at $ns = 7$ while $TAW4TMEFA4$ showed the best results at $steps = 13$. The results indicate that there may be some relation between the history units and the prediction or forecast. However, more research is required to verify this. The

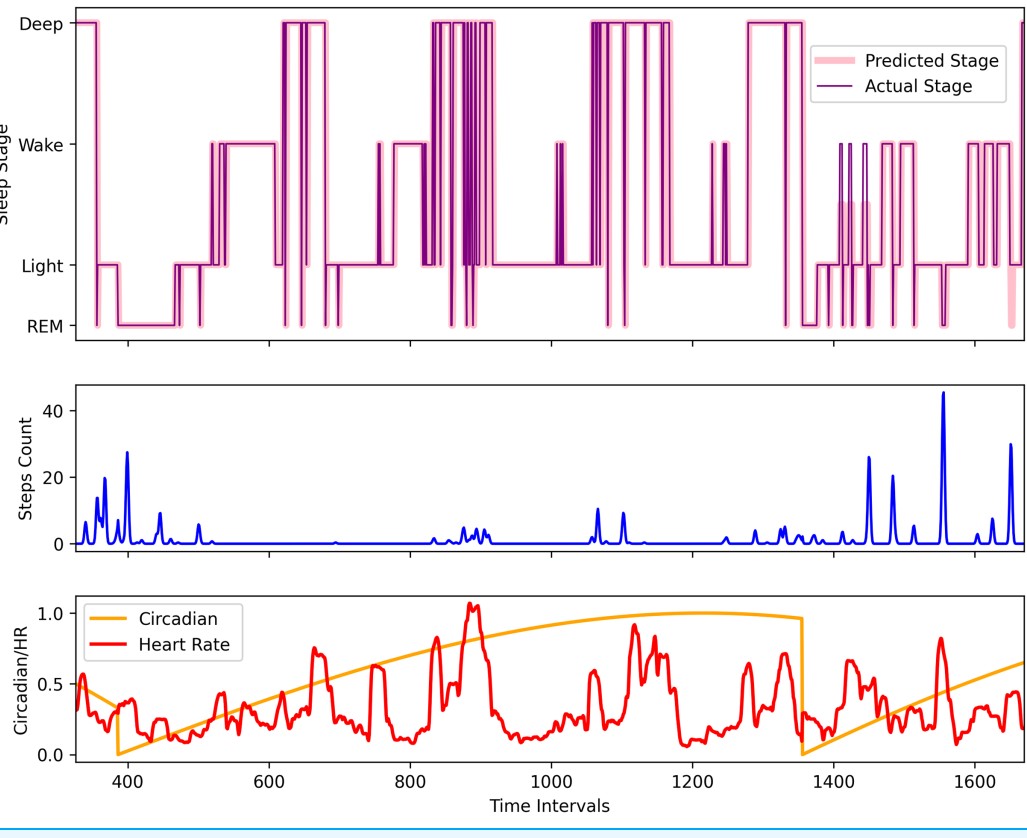

**Figure 10** The comparison of the actual and the predicted sleep pattern for the Apple Watch dataset
**(AW4).**

Fitbit, MESA, and AW datasets are collected with different physical and environmental conditions. Therefore, there is the possibility that the deviation and variance in time units could be the result of this difference. It was also noted that after adding HRV and HR feature to $I1_{act}$ ($= I3_{hrv}$) only ($ns = 1$) was enough to achieve the first maximum accuracy (the same as $AW4$ and $MESA4Act$). It was also interesting to note that $I1_{act}$ had a 60 s interval and the highest accuracy was achieved at $ns = 15$, for $AW$ at $ns = 8$, and $MESA$ at $ns = 1$ (seven unit difference each).

**Machine learning models:** We used weighted evaluation measures for our multiclass problem. Consequently, it can be observed in certain rows of the results that *PRE* is much larger than *ACC*. The reason is that if the algorithm successfully classifies Class 1, and Class 2 but fails to predict Class 3, and Class 4, then *ACC* of the model will be lower, but weighted precision will still be high. However, its macro precision will be less. We have used weighted scores for the evaluation measures.

We have ranked the models according to CK and MC. Algorithms with better CK and *MC* performed well in predicting all classes. RF outclassed all other algorithms in all instances. SVM did not work for the MESA instance, as the dataset was too large and had multiple classes. We experimented with different kernels, standardization, and normalization techniques, and C scores. However, results could not be generated.

## CONCLUSION

In conclusion, this study has made significant strides in leveraging wearable technology for the analysis and prediction of sleep stages and patterns. By employing both memoryless and memory-based models, the research offers insightful findings into the effectiveness of various computational approaches in sleep study.

The Random Forest classifier, representing the memoryless models, emerged as notably proficient, achieving an impressive accuracy (ACC) of 0.96 and a Cohen Kappa score of 0.96. This highlights its superior ability to handle the multifaceted nature of sleep data, outshining other models such as logistic regression, multi-layer perceptron, k-nearest neighbors, and support vector machine. The long short-term memory (LSTM) model, a memory-based model, also demonstrated robust performance, attaining a maximum accuracy of 0.88 and a Kappa score of 0.82 across different datasets. This underscores the significance of incorporating memory elements in models to accurately capture the complexities of sleep patterns. The methodology's efficacy was further validated on diverse datasets, including the Multi-Ethnic Study of Atherosclerosis (MESA) and an Apple Watch dataset from Physio-Net. In the MESA dataset, memoryless models achieved an ACC of 0.75 and Kappa of 0.64, while memory-based models scored 0.86 and 0.78, respectively. For the Apple Watch dataset, the memoryless models recorded an ACC of 0.93 and Kappa of 0.93, compared to 0.92 (ACC) and 0.87 (Kappa) for memory-based models.

These results not only confirm the potential applicability of the proposed models in different sleep data scenarios but also suggest their usefulness in clinical settings for sleep monitoring and analysis. The high accuracy and reliability of these models could assist in the diagnosis and treatment of sleep disorders. Additionally, the study's methodology surpassed existing methods in sleep stage prediction, indicating a significant advancement in the field. This opens new possibilities for the development of sophisticated tools for sleep analysis.

In summary, the integration of advanced computational models with wearable technology is a promising development for understanding sleep and its health implications. Future research should focus on refining these models and exploring their practical applications, ultimately contributing to the field of sleep medicine and enhancing overall health and well-being. The results indicate that a pattern exists within classes as evident from the best models, *i.e.*, RF and kNN where the former works with rule-based classifiers and the latter labels the class with respect to closeness. Similarly, memory-based models indicated the presence of a possible pattern within the transition of sleep stages. This study could be further extended in the classification of abnormal sleep patterns.

## ACKNOWLEDGEMENTS

S. Waqar thanks Mr. Samyan Qayyum Wahla for intellectual discussions and technical support.

### Funding

The authors received no funding for this work.

### Competing Interests

The authors declare that they have no competing interests.

### Author Contributions

- Sahar Waqar conceived and designed the experiments, performed the experiments, analyzed the data, performed the computation work, prepared figures and/or tables, authored or reviewed drafts of the article, and approved the final draft.
- Muhammad Usman Ghani Khan conceived and designed the experiments, authored or reviewed drafts of the article, and approved the final draft.

### Ethics

The following information was supplied relating to ethical approvals (*i.e.*, approving body and any reference numbers):

This research problem is approved by the research committee of the University of Engineering and Technology Lahore, Pakistan.

### Data Availability

The data is available at FigShare: Waqar, Sahar (2023). ECG dataset UET Lahore. figshare. Dataset. https://doi.org/10.6084/m9.figshare.24911613.v1.

The code is available in the Supplemental File.

The third party data of Physionet is available at Walch, O. (2019). Motion and heart rate from a wrist-worn wearable and labeled sleep from polysomnography (version 1.0.0). PhysioNet. https://doi.org/10.13026/hmhs-py35.

Multi-Ethnic Study of Atherosclerosis (MESA) Dataset is available at https://sleepdata.org/datasets/mesa.

### Supplemental Information

Supplemental information for this article can be found online at http://dx.doi.org/10.7717/peerj-cs.1988#supplemental-information.

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
