# Peer review of "Sleep stage prediction using multimodal body network and circadian rhythm"

_PeerJ Computer Science, doi:10.7717/peerj-cs.1988_

## Round 0.1 · original submission · Major Revisions

Check for professional English (editing).

**Language Note:** The Academic Editor has identified that the English language must be improved. PeerJ can provide language editing services - please contact us at copyediting@peerj.com for pricing (be sure to provide your manuscript number and title). Alternatively, you should make your own arrangements to improve the language quality and provide details in your response letter. – PeerJ Staff

Reviewer 1 ·

Basic reporting

Sleep stage prediction is certainly an interesting research question and I like the idea of using simple physiological data collected by wearable devices as the inputs. I do find the intro & background are not too ambiguous to read, the figure quality, however, can be improved further. In Figure 1, the 3D bar plot was very difficult to understand. For MESA, I could barely see the distribution of Deep and Rem as they were hidden in the back. I think authors should just split them into several 2D regular bar plots. For Table 3, in the results (ACC) column what does “86 - 90” mean? Is this confidence interval or standard deviation? It is not entirely clear to me. As for figure 3, authors should just report using bar graphs and grouped by each individual model experimented. Reporting the results in line plots is just misleading. Similarly for figure 6, 7, and 8, the authors should do a better job using different color bars (not just gradient of blues, which makes comparison very difficult) and labeling both axes (figure 8).

Experimental design

Despite the fact that this is an interesting question to study, the experimental design is not clear. It seems that the authors are trying to experiment different ML models on existing datasets and comparing with original performance, but also intended to collect their own datasets. However, I don’t see how they are related. For example, under the methodology section, authors mentioned data was collected from two subjects (1 male and 1 female), but how was this data used in the study? This was not entirely clear to me? In addition, I do want to mention that 2 subjects are too few to generate any results with enough statistical power. Not to mention the question: are these two subjects healthy? The demographics data was not reported either. I think authors should just clearly differentiate the difference between different datasets used and detail the purpose of each.

Validity of the findings

Unfortunately the logic of the entire study was very difficult to follow. It is not clear to me how the evaluation metric aggregates actual sleep stages and predicted sleep stages over all the timestamps (for example, figure 9 in discussion). I do think a significant revision is needed.

Cite this review as

Reviewer 2 ·

Basic reporting

Basically the article is written well with a professional structure. Anyway, the authors can be directed to rewrite to third person.

Experimental design

The study utilized memory-efficient models like LSTM to consider past information in sleep stage predictions. LSTM networks can remember information beyond the recent state, making them suitable for capturing the cyclical patterns in sleep data. The study aimed to find the optimal number of past time instants to consider in sleep stage prediction. The results of the machine learning models, specifically Logistic Regression, Random Forest, k-Nearest Neighbors, Support Vector Machine, and Multi-Layer Perceptron were discussed. It highlights the performance of these models across different database instances and their dependencies on features such as activity and heart rate. It also mentions the use of GridSearchCV to optimize model parameters. In general, the experiments were designed, conducted with sufficient dataset and analyzed well

Validity of the findings

Normally a conclusion serves its purpose of summarizing the research and highlighting its potential implications for future studies and practical applications. Though the summary is clearly and concisely given the implication for future research, research significance, etc. I advise the authors to add the summary of their work, analysis and findings in the conclusion.

Cite this review as

·

Basic reporting

- The manuscript is generally well-written with professional English, but there are a few areas that could be clarified further.
- There are appropriate literature references and the background of the field is provided to contextualize the work.
- The structure of the article is professional and aligns with typical standards for this type of research. However, some concerns arise from specific sections, such as line 255-257 where the transformation of activity data into light information is presented. While the assertion is that "activity is usually carried out under light conditions", it doesn't account for scenarios where there might be inactivity during periods of light or vice versa. A more detailed exposition on the rationale and specific methodology used to accurately derive light information from activity data is essential.
- Figure 9 could benefit from improvement in its presentation. Specifically, the Y-axis labels should represent sleep stages in descriptive terms like "light sleep", "deep sleep", and "REM" instead of integer numbers. Additionally, the height of these figures could be adjusted for better clarity.
- Formal results are generally presented well, but there are instances, such as in the description of dataset features in line 168, where HR was mentioned twice erroneously.

Experimental design

- The research appears to be original and falls within the Aims and Scope of the journal.
- The research question is mostly well-defined, but there is skepticism regarding the sample rate of vital signals and sleep stage labels being every 60 seconds. This raises questions about the sufficiency of the data for sleep stage prediction.
- The methods employed are rigorously described. However, there are points of ambiguity such as in line 259 where the application of the sliding window method needs clarification, especially in terms of labeling.
- The method descriptions could benefit from uniformity in expression. Terms like "time instant i", "time instant t", and "time t" are used interchangeably. It would be beneficial to use "time t" consistently for clarity and consistency.

Validity of the findings

- There is a concern about the dataset only comprising two subjects, which might not offer a diverse demographic representation. However, the time span of the data collection is appreciable.
- The dataset imbalance, with the "light sleep" class being overrepresented compared to the "wake" class, might affect model training and bias predictions. It is essential to discuss if this imbalance has influenced the model's performance.
- The paper commendably carried out extensive work, especially in analyzing the effect of different feature sets and the metrics used in comparisons with other works. However, comparing results sourced from various original papers that use different datasets poses validity concerns. Such comparisons might be influenced by the quality of the datasets used in the original papers, making it challenging to conclude that the proposed method is superior.
- In the discussion section, the usage of weighted precision and recall for multi-class classification is mentioned. This critical information should be introduced earlier, detailing how weights were determined.

Additional comments

- Overall, the paper has undertaken a comprehensive approach, but there are areas of improvement, particularly in data presentation and clarity of method descriptions.
- The concerns about dataset diversity, imbalance, and the validity of comparative analyses should be addressed to strengthen the paper's contributions.
- The manner in which the activity data is transformed into light information requires more in-depth discussion, as the current explanation might be misleading.
- Consistency in terminologies and presentation will further enhance the paper's readability and understanding.

---

## Round 0.2 · accepted · Accept

The reviewer finds that your revised manuscript addresses all the concerns raised in the initial review. Based on the reviewer's comments and my own reading, I have decided to accept the revised manuscript. The article is much improved and meets the high standard required for publication. We appreciate your hard work. Well done.

·

Basic reporting

The revised manuscript has successfully addressed the concerns raised in the initial review. The overall quality of writing is commendable, with clear and unambiguous language used throughout. Adequate literature references provide the necessary background and context for the study. The article structure adheres to professional standards, and figures, tables, and raw data are presented effectively.

Experimental design

The research maintains its originality and aligns well with the journal's Aims and Scope. The manuscript now provides a more detailed explanation, reassuring readers about the sufficiency of data for sleep stage prediction. The methods are described rigorously, and ambiguities, especially regarding the sliding window method, have been clarified. Consistency in terminology usage, a previous point of contention, has been appropriately addressed for clarity and uniformity.

Validity of the findings

The limited representation of the dataset by only two subjects remains a concern; however, the extended time span of data collection is acknowledged. The paper now explicitly addresses the potential impact of dataset imbalance on model training and predictions.

Additional comments

The revised paper has made substantial improvements, effectively addressing previous concerns. The clarity in data presentation and method descriptions has notably strengthened the manuscript. The paper appropriately acknowledges and addresses concerns about dataset diversity, imbalance, and the validity of comparative analyses. Consistency in terminologies and presentation has significantly enhanced the overall readability and understanding of the paper. Overall, the revisions have resulted in a well-polished and improved submission.